# Genetic Polymorphisms of Cytochromes P450 in Finno-Permic Populations of Russia

**DOI:** 10.3390/genes13122353

**Published:** 2022-12-13

**Authors:** Murat Dzhaubermezov, Natalya Ekomasova, Rustam Mustafin, Lilia Gabidullina, Yuliya Galimova, Alfiya Nurgalieva, Yana Valova, Darya Prokofyeva, Elza Khusnutdinova

**Affiliations:** 1Federal State Educational Institution of Higher Education, Ufa University of Science and Technology, 450076 Ufa, Russia; 2Institute of Biochemistry and Genetics, Ufa Federal Research Center, Russian Academy of Sciences, 450054 Ufa, Russia; 3Federal State Educational Institution of Higher Education, Bashkir State Medical University, 450008 Ufa, Russia; 4Federal Institution of Science, Ufa Research Institute of Occupational Health and Human Ecology, 450106 Ufa, Russia

**Keywords:** CYP1A1, CYP2D6, Finno-Permic populations, pharmacogenetic markers

## Abstract

Cytochrome P450 is an enzyme involved in the metabolism of phase 1 xenobiotics, toxins, endogenous hormones, and drugs, including those used in COVID-19 treatment. Cytochrome p450 genes are linked to the pathogenesis of some multifactorial traits and diseases, such as cancer, particularly prostate cancer, colorectal cancer, breast cancer, and cervical cancer. Genotyping was performed on 540 supposedly healthy individuals of 5 Finno-Permic populations from the territories of the European part of the Russian Federation. There was a statistically significant difference between Veps and most of the studied populations in the rs4986774 locus of the CYP2D6 gene; data on the rs3892097 locus of the CYP2D6 gene shows that Izhemsky Komis are different from the Mordovian and Udmurt populations.

## 1. Introduction

The cross-disciplinary integration of the latest research results is particularly important in modern science in order to ensure its effective practical application and reliability. Thus, population genetics studies may contribute to developing new pharmacogenetic approaches, such as tailored treatments targeting representatives of specific ethnic groups with drugs that are more effective for the population in question. The study of polymorphic allelic variants of the cytochrome P450 (CYP450s) genes is of a particular interest to us as its products are responsible for the Phase I biotransformation of drugs of some of the most important categories. CYP450s are heme-containing enzymes that are critical to many cellular processes including eicosanoid metabolism, cholesterol, and bile acid biosynthesis, synthesis, and metabolism of steroids and vitamin D3, biogenic amines formation, and the breakdown and hydroxylation of retinoic acid. There are many other functions of CYP450s including some still not fully understood. Evidence shows that polymorphic CYP450s genes variants are associated with predisposition to such serious diseases as malignant neoplasms (MNs), for example rs1048943 in the *CYP1A1* gene [1].

A study of 100 patients with prostate cancer and 150 healthy people reported an association of the AA rs1048943 genotype with the risk of developing prostate cancer in Iraqi residents [2]. Similar data were obtained regarding the risk of developing colorectal cancer in a study of 200 patients and 200 healthy Iraqis [3]. This is confirmed by meta-analysis of 20 independent original studies involving 8665 patients and 9953 healthy individuals that revealed an association of rs1048943 A > G with an increased risk of developing colorectal cancer [4].

Another meta-analysis caried out in 2016 included 748 patients with laryngeal cancer and a control group of 1558 individuals; it concluded that the G allele and G/G rs1048943 homozygotes in the *CYP1A1* gene were associated with the risk of developing this malignant neoplasm in the Asian population. At the same time, these associated risks were not true for the Caucasian population [1]. This indicates that pharmacogenetic analyses should take ethnicity into account, since such differences, in addition to predisposition to specific neoplasms, may also affect the biotransformation of xenobiotics. In 2018, a meta-analysis of 29 genetic studies showed an increased risk of developing cervical cancer in carriers of the G rs1048943 allele among women from India [5]. In 2021, a meta-analysis using data from 39 studies (7630 patients and 8169 healthy people) reported an association of rs1048943 (AG + GG) in Indians [6].

In 2017, a meta-analysis of 15 studies of breast cancer in Asia (total of 1794 individuals) revealed an association of the rs1065852 *10/*10 (TT) polymorphism in the *CYP2D6* gene with worse disease-free survival and relapse in women receiving adjuvant treatment with tamoxifen dosage of 20 mg/day [7].

In addition, the role of CYP450s isoforms in the metabolism of neurotransmitters, neurosteroids, and cholesterol is currently being studied, as well as their possible effect on behavior including stress, depression, schizophrenia, cognitive processes, learning, and memory. Haduch and Daniel (2019) emphasize the significance of the CYP-mediated alternative pathways for the synthesis of dopamine and serotonin, as they are critical in the local production of these neurotransmitters specifically in the areas of the brain severely affected by depression and schizophrenia, where these neurotransmitter systems are impaired [8]. Most antipsychotics and antidepressants are also metabolized by polymorphic CYP2D6 enzymes and their capacity is genetically determined, which undoubtedly increases the importance of this study [9,10].

The role of cytochrome P450 genes in the pathogenesis of infectious diseases, COVID-19 in particular, is currently being studied, as well as its effect on drug therapy. It was shown that the expression of CYP2D6 decreases in mice infected with hepatitis C virus (HCV) [11], while the expression of CYP2A5 and CYP3A increases in transgenic mice infected with hepatitis B virus (HBV) [12]. Most drugs used to treat COVID-19 are metabolized by cytochrome P450 (CYP) enzymes, primarily CYP2D6 [13]. It was demonstrated that the CYP2D6 variant is associated with the hydroxychloroquine metabolic ratio, which was recently used in COVID-19 treatment [14]. The CYP2D6 and CYP2C19 genes were responsible for most treatment modifications, and the medications most often affected were ondansetron, oxycodone, and clopidogrel, commonly given to patients with COVID-19 [15].

The Finno-Ugric ethno-linguistic community is currently one of the largest language groups in Europe with total of more than 25 million people. The Finno-Ugric branch of languages is divided into two large sub-branches: Finno-Permic and Ugric. Most modern Finno-Ugric languages belong to the Finno-Permic sub-branch while accounting for less than half of the population [16,17].

It is assumed that the ancestral pre-Finno-Ugric population belonged to a single anthropological group originated in ancient Ural. However, modern Finno-Ugric peoples are extremely diverse. Thus, modern Karelians and Vepsians can be described as Caucasians of White Sea-Baltic type, while most of the Mordovians-Erzi are of the Atlanto-Baltic Sura type, and the Mordovians-Mokshas are of the Subural type. The Udmurts belong to the Vyatka-Kama sublaponoid anthropological type, which is characterized by the predominance of Caucasian features over the also present Mongoloid ones.

Population genetics studies of the Finno-Ugric peoples have shown that the modern speakers of these languages and their geographical neighbors are alike in terms of genetic composition of these biological populations. However, when studying connections between geographically distant populations, it is revealed that most of the speakers of these languages and some of their neighbors share a common genetic component, possibly of Siberian origin. In addition, it has been shown that the number of identical IBD segments is much higher among most Uralic-speaking peoples compared to their closest geographic neighbors belonging to other language families [18].

The objective of this research is to study the main pharmacogenetic markers among the Finno-Permic peoples populating the European part of Russian Federation.

## 2. Materials and Methods

The sample used in this study included 540 presumably healthy individuals of 5 Finno-Permic populations from the territories in the European part of the Russian Federation (Figure 1). The sample was divided into 8 groups accounting for the ethnoterritorial distinctions within the populations (Table 1). Sampling was carried out in accordance with the ethical standards of the Bioethics Committee, developed by the WMA Declaration of Helsinki: “Ethical Principles for the Conduct of Medical Research Involving Human Subjects”. All subjects filled out a questionnaire taking into account nationality (up to three generations) and year of birth. All respondents signed an informed voluntary consent to participate in the study. The work was approved by the Local Ethics Committee of the Institute of Biochemistry and Genetics of the USC RAS (protocol No. 14 of 15 September 2016).

The DNA was extracted from peripheral blood samples using phenol-chloroform [19]. Vacutainer^®^ tubes were used to collect, transport, and store the blood samples using 0.5 M EDTA solution as a preservative. After drawing the sample, each tube was shaken and stored at 4 °C.

TaqMan real-time PCR technology was used for the genotyping of polymorphic loci. The incidence of allele variants in given populations were calculated based on observed genotype frequencies. The correspondence of the genotype frequencies to the Hardy–Weinberg equilibrium was assessed using Pearson’s χ^2^ test (at *p* > 0.05). The significance of differences in allele frequencies in the sample was calculated by the χ^2^ test using the Yates correction for continuity. Surfer 24.1.181 was used to produce allele distribution maps.

## 3. Results

In this paper we studied 4 polymorphic loci located in two genes of the cytochrome P450 system (rs1048943, rs1065852, rs3892097, and rs4986774). The distribution of genotype frequencies corresponded to the Hardy–Weinberg equilibrium in most Finno-Permic populations studied by us. Among the few exceptions were the polymorphisms rs1048943, rs1065852, rs3892097 in Mordovian population with *p* < 0.05, as well as the distribution of genotypes of the rs1065852 polymorphic variant of the *CYP2D6* gene in the Udmurt population. Based on this data alone, it is impossible to determine the cause of such a deviation, however, the effect of natural selection on these loci seems to be the most reasonable explanation for this phenomenon.

Paired comparison tests were run for allele frequencies and all selected markers in the studied ethnic groups. In addition, the analysis included data on some other world populations previously published in the academic literature, as well as the data obtained inthe 1000 genomes project [20].

The ethnogenesis of modern Finno-Ugric peoples is an extremely complex topic. The collapse of the ancient Ural community, according to researchers [21,22], occurred in the forth and third millennium BC, when, as R.G. Kuzeev suggests, the tribes of the Keltiminar culture, who came from the south, broke up into two separate groups. The ancestors of the Samoyeds moved to the Yenisei, while the Western groups remained in their former territories. In the third millennium, the latter moved to the west, and mixed with the newcomers formed the Late Neolithic Finno-Ugric community on the territory of the Volga-Kama, Urals, and Trans-Urals [23]. The third and second millennia BC were marked by the continued migration of smaller groups of Finno-Ugric peoples to the north, to the White Sea. The geographical factors and the distance between the tribes led to the final breakup of Finno-Ugric community into Ugric and Finno-Perm groups. Thus, each ethnic group considered in this study developed unique features that are different from related populations due to the timescale and the complexity of its ethnogenesis, the sheer vastness of the populated territories, as well as the relatively large number of neighbors that are different in language and anthropology, which increases the value of this study.

In our study, it was found that the frequency of the genotype AA rs1048943 of the CYP1A1 gene is evenly distributed among all the studied populations and the minimum value is observed in the Udmurt population (88.54%) while the maximum value is in the Erzya subpopulation (96.15%). The GG genotype was found in the Mordovian and Komi populations with frequencies of 1.47% and 1.06%, respectively (Table 2). Table 3 shows the frequency of the minor allele rs1048943 of the CYP1A1 gene in the studied samples of Finno-Permic peoples, as well as in some world populations. Based on the cross-comparison of the populations (*p*-value), it can be concluded that there are no statistically significant differences between allele frequencies in all studied populations. At the same time, the statistically significant differences between the Besermyan and Erzya and the populations of Tatars and Bashkirs, as well as Komi and Bashkirs, are of interest. Even though both Tatars and Bashkirs live in the Volga-Ural region, may be explained by the greater effect of East Eurasian component on the gene pool of the indicated Turkic-speaking populations.

The frequency distribution of the rs1065852 genotypes of the CYP2D6 gene in the studied Finno-Permic populations shows that TT genotype is absent, while the frequency of the minor T allele ranges from 6.38% (95% CI 2.38–13.38) in the Komi population from the Izhma region, and up to 18.23% (95% CI 13.04–24.43) in the Udmurt population (Table 4). The population comparison (*p*-value) revealed complex interrelations within the studied populations (Table 5). Thus, there are statistical differences between the Komi and Mordovian populations, the Moksha subpopulation, and the Udmurts. Komi from the Izhma region, in addition to all the listed populations, have a marked difference from the Erzya population. There is also a significant difference between Veps and Mordovians-Moksha.

The AA homozygotes are also absent in the frequency distribution of the rs3892097 genotypes of the CYP2D6 gene, while the distribution of GA heterozygotes varies considerably from the lowest value of 12.77% in the Komi population of the Izhma region to the maximum in the Mordvin-Moksha population at 37.14% (Table 6). The minor allele A frequency observed in the Komi of the Izhma region is the least and is 6.38% (95% CI 2.38–13.38), while the maximum frequency is 18.57% (95% CI 10.28–29.66) found in the Mordovian-Moksha population. When calculating the significance level (*p*-value), a statistically significant difference was revealed between the Komi population from the Izhemsky district of the Komi Republic and three other populations: the Udmurt populations, the total sample of Mordovians, and with the Mordovian-Moksha subpopulation (Table 7).

It was found that the frequency distributions of alleles and genotypes rs4986774 of the CYP2D6 gene in the studied populations differ significantly (Table 8). In the populations of the Udmurts, Besermens, and two ethnoterritorial groups of the Komi, there was no diversity in the frequencies of genotypes and alleles at all, and the AA genotype is detected in 100% of the samples. In the Karelian population and in the subpopulations of the Mordovians, the frequency of the minor allele (delA) was under 2%, while the Veps population is fundamentally different, and the minor allele frequency is 6.67% (95% CI 2.92–12.71). This difference in allele frequencies may be explained by drift and/or the founder effect. When calculating the *p*-value, a statistically significant difference was revealed between the Veps population and all the other populations in the sample, except for Karelians and Moksha-Mordovians. (Table 9). Due to the lack of rs4986774 of the *CYP2D6* gene in the databases, only the samples analyzed in this study were compared in Table 9.

## 4. Discussion

Based on the results of our study, as well as literature data, we built maps of the frequency distribution of minor alleles of the studied loci in the Surfer program (Figure 2).

The frequency distribution of alleles and genotypes of the rs1048943 polymorphic variant in various populations of the world has already been described. Thus, the maximum values of the 462Val variant were found in the indigenous peoples of North and South America (over 70%), as well as in the populations of East Asia (over 30%) and Kazakhstan (28.4%) [26,27,28,29,30]. The lowest values are typical for the population of Africa, Europe, and some populations of Western Asia (0–3%, 2–7% and 5.8–9.5%, respectively) [2,20,24,31,32].

In this paper, the polymorphism of the Phase I gene of the CYP1A1 xenobiotic biotransformation system (A2455G, rs1048943) was studied for the first time in the Finno-Permic populations inhabiting the vast territories of the European part of the Russian Federation. The data obtained show no significant differences between the Finno-Permic populations included in the study, with the exception of Turkic-speaking populations of the Bashkirs and Tatars. In the studied populations, the *CYP1A1* (2455G) allele occurs with frequencies typical for Western Asian and European populations. These new data on the xenobiotic biotransformation genes allele frequency distribution should be considered in further pharmacogenetic studies.

Cytochrome P450 2D6 is involved in the metabolism of many drugs including those used in treatment of cancer and cardiac diseases [33]. Our study shows that the frequencies of genotypes and alleles rs1065852 of the *CYP2D6* gene obtained for our sample of the Komi, Komi of the Izhma region, Veps, and Karelians are statistically different from the typical pan-European frequency values. The populations with remarkably similar results are of a particular interest, especially the Karelians and the Finn population [20], which are closely related culturally, linguistically, and geographically.

## 5. Conclusions

A worldwide study of rs1048943 in the CYP1A1 gene revealed an association of the G allele with prostate cancer [2] and colorectal cancer in Iraqi population [3]. Meta-analyses have shown this allele to be associated with colorectal cancer regardless of the ethnic origin [4], with laryngeal cancer in Asians (but not in Caucasians) [1], cervical cancer [5], and lung cancer in Indians [6]. The rs1065852 (TT) in the CYP2D6 gene is reported to be associated with worse prognosis for breast cancer in Asian women [7]. Thus, rs1048943 allele variants in the CYP1A1 gene represent the risk of developing the most common malignancies, both regardless of population origin (colorectal cancer) and for certain regions and specific ethnic groups. In this regard, it is relevant to study the distribution of alleles in different populations, which can aid early diagnostics and more accurately predict risks related to specific cancers. It can be assumed that such a distribution of alleles affects not only the risk of developing cancer, but also the effectiveness of chemotherapy and, therefore, the survival of patients. With further comparative studies of the malignant neoplasms in these populations, it is possible to predict the development of tumors and select the best pharmacotherapy for specific groups of patients.

It is also important that a large number of antipsychotics and antidepressants, as well as drugs used for COVID-19 treatment, are metabolized by polymorphic CYP2D6 enzymes. Thus, understanding the distribution of CYP2D6 alleles in the Finno-Permic populations will allow us to develop new approaches to treatment targeting particular ethnic groups.

This study of the main pharmacogenetic markers shows statistically significant difference between Veps and most of the studied populations in the rs4986774 locus of the CYP2D6 gene; as for the rs3892097 locus of the CYP2D6 gene, here we can see that Izhemsky Komis are different from the Mordovian and Udmurt populations.

## Figures and Tables

**Figure 1 genes-13-02353-f001:**
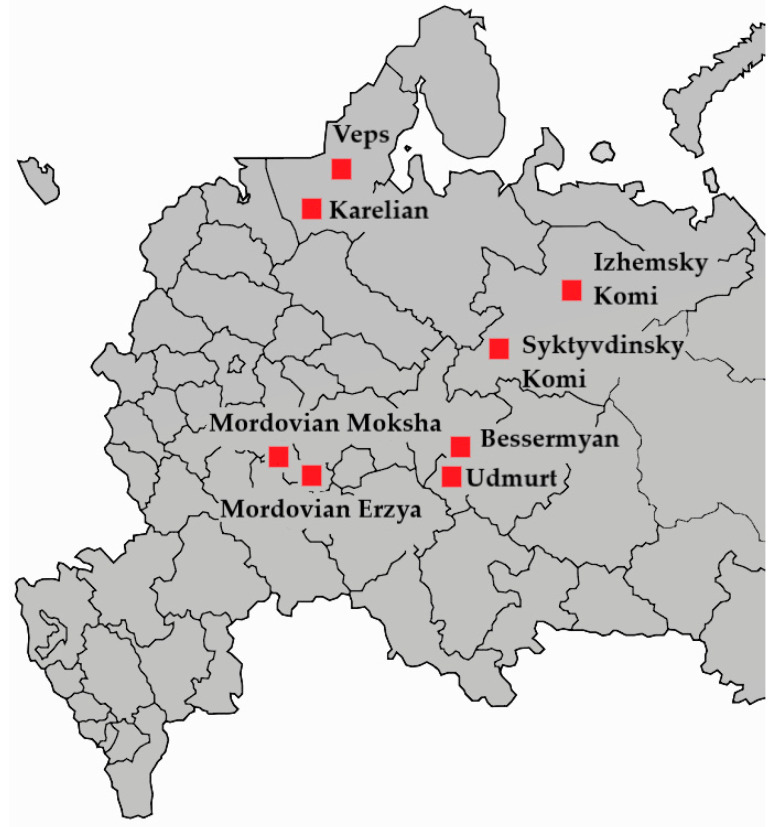
Map of the European part of Russia indicating the places of biomaterial collection.

**Figure 2 genes-13-02353-f002:**
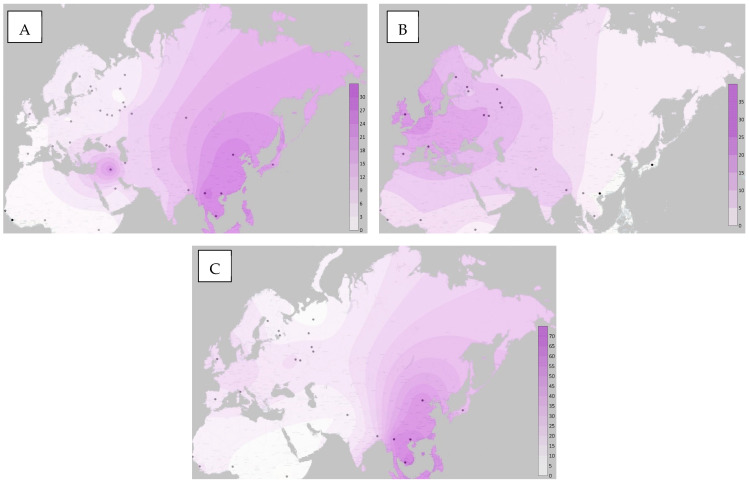
Distribution maps of minor alleles (**A**) rs1048943, (**B**) rs3892097, (**C**) rs1065852.

**Table 1 genes-13-02353-t001:** Linguistic picture of the Finno-Permic population of Russia, studied in the work, as well as the amount of material used.

Sub-Branch of Finno-Permic Languages	Language Group	Language Sub-Group	Populations Included in the Study	N
Finno-Permic sub-branch	Permic	-	Udmurt	96
			Besermyan	94
			Komi	94
	Finno-Volga	Mordovian	Erzya	78
			Moksha	35
		Baltic-Finnish	Karelian	60
			Veps	60

**Table 2 genes-13-02353-t002:** Distribution of the rs1048943 genotypes of the CYP1A1 gene in the Finno-Permic populations of the Russian Federation.

Population	N	AA			AG			GG			Minor Allele Frequency (95% CI)	χ^2^	Deviations from HWE, *p*
		Observed (N)	Expected (N)	%	Observed (N)	Expected (N)	%	Observed (N)	Expected (N)	%			
Veps	60	55	55.1	91.67	5	4.8	8.33	0	0.1	0	4.17% (1.36–9.45)	0.113	0.736
Karelian	60	55	55.1	91.67	5	4.8	8.33	0	0.1	0	4.17% (1.36–9.45)	0.113	0.736
Udmurt	96	85	85.3	88.54	11	10.4	11.46	0	0.3	0	5.73% (2.89–10.02)	0.354	0.551
Bessermyan	94	90	90	95.74	4	3.9	4.26	0	0	0	2.13% (0.58–5.36)	0.044	0.833
Mordovian (total)	136	128	126.2	94.12	6	9.6	4.41	2	0.2	1.47	3.68% (1.78–6.66)	19.340	0.00001
Erzya	78	75	75	96.15	3	2.9	3.85	0	0	0	1.92% (0.40–5.52)	0.03	0.862
Moksha	35	32	31.1	91.43	2	3.8	5.71	1	0.1	2.86	5.71% (1.58–13.99)	7.721	0.005
Komi (total)	94	89	88.1	94.68	4	5.8	4.26	1	0.1	1.06	3.19% (1.18–6.82)	9.113	0.002
Izhemsk	47	44	44	93.62	3	2.9	6.38	0	0	0	3.19% (0.66–9.04)	0.051	0.821
Syktyvdinsk	47	45	44	95.74	1	2.9	2.13	1	0	2.13	3.19% (0.66–9.04)	20.20	0.000007

**Table 3 genes-13-02353-t003:** Frequencies of the minor allele rs1048943 of the *CYP1A1* gene in the studied samples of Finno-Permic peoples, as well as in some populations of the world.

Population	N	Minor allele Frequency%	Veps	Karelian	Mordovian (Total)	Erzya	Moksha	Udmurt	Komi (Total)	Izhemsk	Syktyvdinsk	Bessermyan
Veps	60	4.17%		0.747	0.958	0.459	0.896	0.730	0.893	0.992	0.992	0.491
Karelian	60	4.17%	0.747		0.958	0.459	0.896	0.730	0.893	0.992	0.992	0.491
Mordovian (total)	136	3.68%	0.958	0.958				0.412	0.984	0.917	0.917	0.500
Erzya	78	1.92%	0.460	0.460			0.269	0.128	0.693	0.835	0.835	0.803
Moksha	35	5.71%	0.896	0.896		0.269		0.767	0.568	0.689	0.689	0.283
Udmurt	96	5.73%	0.730	0.730	0.418	0.128	0.767		0.343	0.520	0.520	0.124
Komi (total)	94	3.19%	0.893	0.893	0.984	0.693	0.568	0.343				0.749
Izhensk	47	3.19%	0.992	0.992	0.917	0.835	0.689	0.520			0.678	0.892
Syktyvdinsk	47	3.19%	0.992	0.992	0.917	0.835	0.689	0.520		0.678		0.892
Bessermyan	94	2.13%	0.491	0.491	0.500	0.803	0.283	0.124	0.749	0.892	0.892	
Finnish [20]	99	5.1%	0.930	0.930	0.619	0.204	0.923	0.942	0.509	0.677	0.677	0.207
Russian [24]	314	4.8%	0.957	0.957	0.576	0.172	0.958	0.733	0.468	0.673	0.673	0.165
Tatar, Russia [24]	243	6.4%	0.482	0.482	0.158	**0.050**	0.961	0.889	0.150	0.335	0.335	**0.041**
Bashkir, Russia [24]	134	10.5%	0.063	0.063	**0.004**	**0.002**	0.329	0.105	**0.006**	0.051	0.051	**0.001**
England and Scotland [20]	91	3.3%	0.935	0.935	0.964	0.658	0.603	0.379	0.813	0.756	0.756	0.709
Spain [20]	107	1.9%	0.372	0.372	0.363	0.727	0.203	0.073	0.597	0.763	0.763	0.863
Toscana, Italy [20]	107	3.3%	0.908	0.908	0.994	0.642	0.573	0.337	0.812	0.754	0.754	0.693
Balkar [25]	104	8.6%	0.191	0.191	**0.035**	**0.012**	0.595	0.350	**0.039**	0.138	0.138	**0.009**
Karachay [25]	73	7.5%	0.373	0.373	0.137	**0.040**	0.837	0.657	0.125	0.263	0.263	**0.035**
Europe [20]	503	3.5%	0.901	0.901	0.977	0.438	0.524	0.199	0..984	0.881	0.881	0.463
S. Asia [20]	489	12.7%	**0.006**	**0.006**	**0.00002**	**0.00008**	0.126	**0.008**	**0.0001**	**0.011**	**0.011**	**0.00002**

Bold indicates statistically significant differences (*p* < 0.05).

**Table 4 genes-13-02353-t004:** Distribution of the rs1065852 genotypes of the *CYP2D6* gene in the Finno-Permic populations of the Russian Federation.

Population	N	CC			CT			TT			Minor Allele Frequency (95% CI)	χ^2^	Deviations from HWE, *p*
		Observed (N)	Expected (N)	%	Observed (N)	Expected (N)	%	Observed (N)	Expected (N)	%			
Veps	60	47	47.7	78.33	13	11.6	21.67	0	0.7	0	10.83% (5.90–17.81)	0.886	0.347
Karelian	60	45	45.9	75	15	13.1	25	0	0.9	0	12.50% (7.17–19.78)	1.224	0.268
Udmurt	96	61	64.2	63.54	35	28.6	36.46	0	3.2	0	18.23% (13.04–24.43)	4.771	0.029
Bessermyan	94	64	66.4	68.09	30	25.2	31.91	0	2.4	0	15.96% (11.03–21.99)	3.389	0.066
Mordovian (total)	136	87	91.4	63.97	49	40.2	36.03	0	4.4	0	18.01% (13.63–23.11)	6.566	0.01
Erzya	78	52	54.2	66.67	26	21.7	33.33	0	2.2	0	16.67% (11.19–23.46)	3.12	0.077
Moksha	35	19	20.8	54.29	16	12.3	45.71	0	1.8	0	22.86% (13.67–34.45)	3.073	0.08
Komi (total)	94	75	76	79.79	19	17.1	20.21	0	1	0	10.11% (6.20–15.33)	1.188	0.276
Izhemsk	47	41	41.2	87.23	6	5.6	12.77	0	0.2	0	6.38% (2.38–13.38)	0.218	0.640
Syktyvdinsk	47	34	34.9	72.34	13	11.2	27.66	0	0.9	0	13.83% (7.57–22.49)	1.211	0.271

**Table 5 genes-13-02353-t005:** Frequencies of the minor allele rs1065852 of the *CYP2D6* gene in the studied samples of Finno-Permic peoples, as well as in some populations of the world.

Population	N	Minor allele Frequency%	Veps	Karelian	Mordovian (Total)	Erzya	Moksha	Udmurt	Komi (Total)	Izhemsk	Syktyvdinsk	Bessermyan
Veps	60	10.83%		0.841	0.100	0.228	**0.044**	0.109	0.990	0.371	0.691	0.273
Karelian	60	12.50%	0.841		0.225	0.427	0.097	0.237	0.640	0.207	0.982	0.501
Mordovian (total)	136	18.01%	0.100	0.225				0.950	**0.027**	**0.011**	0.396	0.653
Erzya	78	16.67%	0.228	0.427			0.357	0.811	0.102	**0.031**	0.626	0.975
Moksha	35	22.86%	**0.044**	0.097		0.357		0.509	**0.014**	**0.005**	0.176	0.269
Udmurt	96	18.23%	0.109	0.237	0.950	0.811	0.509		**0.034**	**0.012**	0.402	0.651
Komi (total)	94	10.11%	0.990	0.640	**0.027**	0.102	**0.014**	**0.034**				0.125
Izhemsk	47	6.38%	0.371	0.207	**0.011**	**0.031**	**0.005**	**0.012**			0.161	
Syktyvdinsk	47	13.83%	0.691	0.982	0.396	0.626	0.176	0.402		0.161		0.717
Bessermyan	94	15.96%	0.273	0.501	0.653	0.975	0.269	0.651	0.125	**0.037**	0.717	
Finnish [20]	99	14.6%	0.422	0.712	0.399	0.709	0.163	0.413	0.231	0.066	0.940	0.829
England and Scotland [20]	91	24.7%	**0.004**	**0.014**	0.107	0.093	0.884	0.160	**0.0003**	**0.0004**	**0.043**	**0.049**
Spain [20]	107	17.3%	0.154	0.317	0.930	0.986	0.389	0.907	0.053	**0.018**	0.508	0.823
Toscana, Italy [20]	107	20.6%	**0.034**	0.088	0.554	0.418	0.810	0.640	**0.006**	**0.003**	0.188	0.289
Europe [20]	503	20.2%	**0.014**	**0.044**	0.426	0.305	0.700	0.535	**0.001**	**0.002**	0.152	0.180
S. Asia [20]	489	16.5%	0.111	0.264	0.545	0.949	0.225	0.549	**0.027**	**0.015**	0.551	0.864

Bold indicates statistically significant differences (*p* < 0.05).

**Table 6 genes-13-02353-t006:** Distribution of the rs3892097 genotypes of the *CYP2D6* gene in the Finno-Permic populations of the Russian Federation.

Population	N	GG			GA			AA			Minor Allele Frequency (95% CI)	χ^2^	Deviations from HWE, *p*
		Observed (N)	Expected (N)	%	Observed (N)	Expected (N)	%	Observed (N)	Expected (N)	%			
Veps	60	48	48.6	80	12	10.8	20	0	0.6	0	10% (5.27–16.82)	0.741	0.389
Karelian	60	43	44.2	71.67	17	14.6	28.33	0	1.2	0	14.17% (8.47–21.71)	1.634	0.201
Udmurt	96	65	67.5	67.71	31	26	32.29	0	2.5	0	16.15% (11.24–22.13)	3.559	0.059
Bessermyan	94	68	69.8	72.34	26	22.4	27.66	0	1.8	0	13.83% (9.24–19.60)	2.421	0.119
Mordovian (total)	136	94	97.2	69.12	42	35.5	30.88	0	3.2	0	15.44% (11.36–20.29)	4.535	0.033
Erzya	78	55	56.7	70.51	23	19.6	29.49	0	1.7	0	14.74% (9.58–21.30)	2.333	0.127
Moksha	35	22	23.2	62.86	13	10.6	37.14	0	1.2	0	18.57% (10.28–29.66)	1.821	0.177
Komi (total)	94	74	75.1	78.72	20	17.9	21.28	0	1.1	0	10.64% (6.62–15.95)	1.332	0.248
Izhemsk	47	41	41.2	87.23	6	5.6	12.77	0	0.2	0	6.38% (2.38–13.38)	0.218	0.640
Syktyvdinsk	47	33	34	70.21	14	11.9	29.79	0	1	0	14.89% (8.39–23.72)	1.439	0.230

**Table 7 genes-13-02353-t007:** Frequencies of the minor allele rs3892097 of the CYP2D6 gene in the studied samples of Finno-Permic peoples, as well as in some populations of the world.

Population	N	Minor allele Frequency%	Veps	Karelian	Mordovian (Total)	Erzya	Moksha	Udmurt	Komi (Total)	Izhemsk	Syktyvdinsk	Bessermyan
Veps	60	10%		0.428	0.200	0.321	0.143	0.173	0.990	0.485	0.380	0.413
Karelian	60	14.17%	0.428		0.863	0.970	0.550	0.756	0.454	0.109	0.963	0.932
Mordovian (total)	136	15.44%	0.200	0.863				0.939	0.179	**0.038**	0.969	0.730
Erzya	78	14.74%	0.321	0.970			0.596	0.833	0.326	0.072	0.880	0.931
Moksha	35	18.57%	0.143	0.550		0.596		0.781	0.137	**0.030**	0.678	0.453
Udmurt	96	16.15%	0.173	0.756	0.939	0.833	0.781		0.154	**0.033**	0.920	0.625
Komi (total)	94	10.64%	0.990	0.454	0.179	0.326	0.137	0.154				0.431
Izhemsk	47	6.38%	0.485	0.109	**0.038**	0.072	**0.030**	**0.033**			0.098	0.097
Syktyvdinsk	47	14.89%	0.380	0.963	0.969	0.880	0.678	0.920		0.098		0.952
Bessermyan	94	13.83%	0.413	0.932	0.730	0.931	0.453	0.625	0.431	0.097	0.952	
Finnish [20]	99	13.6%	0.434	0.972	0.679	0.886	0.423	0.580	0.456	0.103	0.913	0.926
England and Scotland [20]	91	24.2%	**0.003**	**0.048**	**0.027**	**0.042**	0.433	0.070	**0.0009**	**0.0005**	0.101	**0.016**
Spain [20]	107	14.5%	0.315	0.934	0.869	0.936	0.529	0.744	0.314	0.068	0.935	0.964
Toscana, Italy [20]	107	18.7%	0.051	0.366	0.408	0.391	0.877	0.587	**0.033**	**0.009**	0.519	0.239
Europe [20]	503	18.6%	**0.019**	0.234	0.230	0.245	0.877	0.421	**0.008**	**0.005**	0.455	0.118
S. Asia [20]	489	10.9%	0.754	0.292	**0.043**	0.166	0.081	**0.040**	0.903	0.231	0.324	0.310

Bold indicates statistically significant differences (*p* < 0.05).

**Table 8 genes-13-02353-t008:** Distribution of the rs4986774 genotypes of the CYP2D6 gene in the Finno-Permic populations of the Russian Federation.

Population	N	A/A			A/del			del/del			Minor Allele Frequency (95% CI)	χ^2^	Deviations from HWE, *p*
		Observed (N)	Expected (N)	%	Observed (N)	Expected (N)	%	Observed (N)	Expected (N)	%			
Veps	60	52	52.3	86.67	8	7.5	13.33	0	0.3	0	6.67% (2.92–12.71)	0.306	0.580
Karelian	60	58	58	96.67	2	2	3.33	0	0	0	1.67% (0.20–5.89)	0.017	0.895
Udmurt	96	96	96	100	0	0	0	0	0	0			
Bessermyan	94	94	94	100	0	0	0	0	0	0			
Mordovian (total)	136	132	132	97.06	4	3.9	2.94	0	0	0	1.47% (0.40–3.72)	0.03	0.861
Erzya	78	76	76	97.44	2	2	2.56	0	0	0	1.28% (0.16–4.55)	0.013	0.908
Moksha	35	33	33	94.29	2	1.9	5.71	0	0	0	1.99% (0.35–9.94)	0.03	0.862
Komi (total)	94	94	94	100	0	0	0	0	0	0			
Izhemsk	47	47	47	100	0	0	0	0	0	0			
Syktyvdinsk	47	47	47	100	0	0	0	0	0	0			

**Table 9 genes-13-02353-t009:** Frequencies of the minor allele rs4986774 of the *CYP2D6* gene in the studied samples of Finno-Permic peoples.

Population	N	Minor allele Frequency%	Veps	Karelian	Mordovian (Total)	Erzya	Moksha	Udmurt	Komi (Total)	Izhemsk	Syktyvdinsk	Bessermyan
Veps	60	6.67%		0.106	**0.015**	**0.040**	0.425	**0.0008**	**0.0009**	**0.030**	**0.030**	**0.0009**
Karelian	60	1.67%	0.106		0.763	0.808	0.978	0.214	0.223	0.640	0.640	0.223
Mordovian (total)	136	1.47%	**0.015**	0.763				0.272	0.282	0.734	0.734	0.282
Erzya	78	1.28%	**0.040**	0.808			0.776	0.346	0.358	0.842	0.842	0.358
Moksha	35	1.99%	0.425	0.978		0.776		0.053	0.057	0.293	0.293	0.057
Udmurt	96	0%	**0.0008**	0.214	0.272	0.346	0.053					
Komi (total)	94	0%	**0.0009**	0.223	0.282	0.358	0.057					
Izhemsk	47	0%	**0.030**	0.640	0.734	0.842	0.293					
Syktyvdinsk	47	0%	**0.030**	0.640	0.734	0.842	0.293					
Bessermyan	94	0%	**0.0009**	0.223	0.282	0.358	0.057					

Bold indicates statistically significant differences (*p* < 0.05).

## Data Availability

Not applicable.

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
