# Peer review of "Genetic Polymorphisms of Cytochromes P450 in Finno-Permic Populations of Russia"

_genes, 2022, doi:10.3390/genes13122353_

Round 1

Reviewer 1 Report

It is a showpiece study. It is well-performed, well-written, and easy to read. The results will be of interest to a broad range of researchers in the areas of population genetics and pharmacogenomics. In my view, minor corrections and modifications as outlined below:

1.      The final sentence of the abstract (“The study of the key pharmacogenetic markers of the CYP1A1 (rs1048943) and CYP2D6 (rs1065852, rs3892097 and rs4986774) genes did not reveal significant genetic differences between the Finno-Permic populations of the Russian Federation”) and the similar statement in the Conclusion (lines 285-286) sound quite discouraging. However, they may not reflect the results of the study adequately. Indeed, as described in lines 218-219, the authors revealed significant differences in the distribution of the CYP2D6 rs4986774 allele between the studied ethnolinguistic groups. Notable differences in the frequency distributions of the rs3892097 genotypes of the CYP2D6 were also detected (lines 197-207). It is unclear why the authors did not highlight these findings in the abstract and Conclusion. Doing so would increase the scientific soundness of their article.

2.      Figure 1. Map of the European part of Russia indicating the places of biomaterial collection. – I think that the informative value of this figure would benefit from specifying the ethnolinguistic groups analyzed at each locus pointed out in the map.

3.      Line 128 – add “the” before “polymorphisms”

4.      Line 136 – replace “data from” with “the data obtained in”

5.      Line 160 – insert a comma after “(p-value)”

6.      Lines 218-219: “It was found that the frequency distribution of alleles and genotypes rs4986774 of the CYP2D6 gene in the studied populations was uneven (Table 8).” – In this sentence, the word “distribution” should be in the plural (distributions). In addition, “uneven” does not sound like the right word in this context. I would suggest replacing “were uneven” with “differ significantly”.

7.      Line 226: Add a comma after “p-value”.

8.      Line 228:  The word “exclusion” does not sound right in this context. I would suggest replacing it with “omission” or “lack” (in the latter case, “from the databases” should become “in the databases”). In the same sentence, I would replace “the studied samples” with “the samples analyzed in this study” for better clarity.

Author Response

Пожалуйста, посмотрите приложение.

Reviewer 2 Report

Cytochrome P450 enzymes are an important class of monooxygenase that is involved in xenobiotics and drug metabolism. Chemical carcinogen bioactivation by cytochrome P450 enzymes has been implicated in tumorigenesis and some other diseases. Therefore, pharmacogenetics of CYP enzymes is of vital importance for understanding drug metabolism, drug-drug interaction, epidemiology, and etiology of diseases such as cancer.

Your current study focused on the polymorphism of CYP1A1 and CYP2D6, both have been implicated in a variety of public health issues, among the Finn-Permic population in Russia is worthy of commendation. The results are an important addition to our knowledge of CYP pharmacogenetics and would benefit the health care of not only these ethnic groups but also mankind as a whole.

The manuscript is well-written and the data is presented logically and clearly. And the research method was sound and rigorous.

Congratulation!

Author Response

We would like to express our gratitude for you positive review.